# Disruption of Human Papillomavirus 16 E6/E7 Genes Using All-in-One Adenovirus Vectors Expressing Eight Double-Nicking Guide RNAs

**DOI:** 10.3390/ijms26178685

**Published:** 2025-09-05

**Authors:** Megumi Yamaji, Tomomi Nakahara, Tomoko Nakanishi, Satomi Aoyama-Kikawa, Kiyoshi Yamaguchi, Yoichi Furukawa, Mariko Nakamura, Tadashi Okada, Hirotaka Tabata, Ryoko Fuse, Eigo Shimizu, Rika Kasajima, Seiya Imoto, Iwao Kukimoto, Izumu Saito, Tohru Kiyono

**Affiliations:** 1Center of Biomedical Research Resources, Juntendo University School of Medicine, Tokyo 113-8421, Japan; m.yamaji.bj@juntendo.ac.jp (M.Y.); nakanishi-t@juntendo.ac.jp (T.N.); m.nakamura.ou@juntendo.ac.jp (M.N.); 2National Cancer Center Research Institute, Tokyo 466-0045, Japan; tnakahara@biken.osaka-u.ac.jp (T.N.); aoyama.satomi.mf@mail.hosp.go.jp (S.A.-K.); 3Division of Clinical Genome Research, The Institute of Medical Science, The University of Tokyo, Tokyo 108-8639, Japan; kiyamagu@g.ecc.u-tokyo.ac.jp (K.Y.); yofurukawa@g.ecc.u-tokyo.ac.jp (Y.F.); 4Department of Pharmaceutical Engineering, Faculty of Engineering, Toyama Prefectural University, Toyama 939-0398, Japan; okada@owc.ac.jp (T.O.);; 5Laboratory of Virology, Institute of Microbial Chemistry (BIKAKEN), Microbial Chemistry Foundation, Tokyo 141-0021, Japan; 6Division of Health Medical Intelligence, Human Genome Center, The Institute of Medical Science, The University of Tokyo, Tokyo 108-8639, Japan; eigos@hgc.jp (E.S.); rkasajima@gancen.asahi.yokohama.jp (R.K.); imoto@hgc.jp (S.I.); 7Molecular Pathology and Genetics Division, Kanagawa Cancer Center Research Institute, Yokohama 241-8518, Japan; 8Pathogen Genomics Center, National Institute of Infectious Diseases, Tokyo 208-0011, Japan; ikuki@niid.go.jp; 9Department of Physiology, Juntendo University Graduate School of Medicine, Tokyo 113-8421, Japan; 10Project for Prevention of HPV-Related Cancer, Division of Collaborative Research and Development, Exploratory Oncology Research and Clinical Trial Center, National Cancer Center, Chiba 277-8577, Japan; 11Department of Cancer Cell Research, Sasaki Institute, Tokyo 101-0062, Japan

**Keywords:** human papillomavirus, cancer therapy, CRISPR/Cas9, nickase, genome editing, adenovirus vector, all-in-one, multiplex guide RNA, gene therapy

## Abstract

Human papillomavirus (HPV) is a prime target for genome-editing therapy as its E6 and E7 oncogenes are crucial for cancer development and maintenance. A key challenge in CRISPR/Cas9 therapy is the off-target effects. This study utilized a double-nicking technique to introduce DNA breaks in the E6 and E7 regions of HPV16. From 146 gRNA candidates, 16 double-nicking pairs were selected. Multiple combinations of double-nicking (DN)-gRNA pairs were delivered to HPV16-positive cells via lentiviruses, followed by Cas9 nickase (Cas9n) expression. Combinations of 3–4 DN-gRNA pairs effectively killed HPV16-positive cells while sparing HPV-negative cells. Off-target effects were reduced by nearly three orders of magnitude. An “all-in-one” adenovirus (AdV) system expressing four gRNA pairs and Cas9n showed promise in inhibiting tumor growth in HPV16-positive cancer models, demonstrating its potential as a safe and effective treatment for HPV-induced tumors.

## 1. Introduction

Oncogenic human papillomavirus (HPV) infection accounts for approximately 5% of all cancers worldwide, including cervical cancer and various other anogenital and oropharyngeal cancers. In 2020, there were an estimated 600,000 new cases of cervical cancer and 340,000 related deaths globally [1]. Although the neoplastic progression of HPV-associated precursor lesions, as well as the maintenance of established cancers, is driven by the expression of the viral oncogenes E6 and E7 [2,3], no targeted therapies against HPV have been developed. Current standard treatments for cervical and oropharyngeal cancers include surgical resection and/or chemotherapy and radiotherapy, which can significantly diminish quality of life (QOL). These treatments often result in adverse outcomes, such as loss of fertility and long-lasting dyslalia and/or dysphagia [4,5]. Therefore, there is an urgent need to develop non-invasive therapies that preserve QOL post-treatment. Immunotherapies such as immune checkpoint inhibitors, as well as intrabodies or small-molecule drugs targeting the E6 and E7 proteins, are currently under development [6]. In this study, we aimed to develop a novel therapeutic approach for HPV-associated malignancies by introducing CRISPR/Cas9-generated DNA breaks into HPV genomes using an adenovirus vector (AdV).

Advancements in double-nicking technology have demonstrated the potential to reduce off-target effects by up to 1500-fold [7]. This strategy employs Cas9n and two guide RNAs (gRNAs) to introduce nicks in the DNA, with an optimal “offset” of −4 to 20 nucleotides between the 5′ termini of the gRNAs or with 5′-overhangs of 30–54 nucleotides for effective cleavage [7]. The immediate repair of off-target nicks by Cas9n significantly reduces the likelihood of generating double-stranded breaks at unintended sites [8]. However, the double-nicking strategy has not gained widespread use, primarily due to the requirement of two gRNAs for each cleavage event. Additionally, multiplex gRNA expression units can lead to loss of the units during homologous recombination in both Escherichia coli and human cells during vector construction and viral amplification.

To effectively eliminate HPV oncogene expression using CRISPR/Cas9 technology, it is ideal to induce multiple DNA breaks by employing several gRNAs that target different sequences within the HPV genome. In our research, we selected optimal combinations of gRNAs targeting the HPV16 genome to effectively kill HPV16-positive cancer cells in conjunction with Cas9n. Our findings revealed that combinations of three to four pairs of double-nicking gRNAs (DN-gRNAs) targeting the HPV16 genome efficiently induced cell death and resulted in irreversible large deletions of HPV oncogenes. Moreover, the double-nicking strategy reduced off-target insert-deletion (indel) formation to 1/860 compared to wild-type Cas9. Building on our recent methodology [9], we constructed an “all-in-one” AdV that co-expressed four pairs of DN-gRNAs targeting HPV16 along with Cas9n, successfully inhibiting the growth of HPV16-positive tumors. This study provides proof of concept that the combination of Cas9n and multiplex DN-gRNAs may offer a safe and effective therapeutic strategy for HPV-positive cancers.

## 2. Results

### 2.1. In Vitro Cleavage of HPV16 DNA by Cas9, tracrRNA, and crRNAs

Among more than 100 candidate target sites, we manually selected 39 crRNAs and tested their ability to cleave HPV16 DNA segments in vitro. Most of the crRNAs efficiently digested the DNA at the expected sites (Figure 1), while a few crRNAs, such as AS869G, S884G, and AS1192g+, exhibited lower activity in two independent experiments (indicated by star marks). These findings suggest that DN-gRNA sets, including AS507G/S563g+, AS706g+/S753G, and AS859G/S893G, are promising candidates for further investigation using the double-nicking strategy.

### 2.2. Growth Inhibition by gRNAs and Cas9n

Previous studies have demonstrated that targeting the HPV genome with Cas9 and specific gRNAs effectively eliminates HPV-positive cancer cells [10,11,12,13]. To assess whether gRNAs targeting the HPV16 genome, in combination with Cas9n, are effective, we designed gRNAs targeting the E6/E7 region of HPV16 (Figure 2a and Table 1). These gRNAs, along with a control gRNA targeting HPV18 (Appendix A), were delivered to the HPV16-positive SiHa cervical cancer cell line via retrovirus-mediated gene transfer. The cells were then infected with an AdV expressing either wild-type Cas9 (AxCB-Cas9) or Cas9n (AxCB-Cas9n) under the CB promoter [14]. As expected, gRNAs such as S490G and S753G significantly inhibited SiHa cell growth when combined with Cas9, and moderately inhibited growth with Cas9n. Combinations of DN-gRNAs, such as AS507G/S563g, AS507G/S563G+, and AS859G/S893G, significantly inhibited SiHa cell growth with Cas9n, with more potent inhibition observed with wild-type Cas9. However, combinations like AS706g/S753G had similar inhibitory effects to the single gRNA S753G. Combinations like AS57G/S107gA+ and AS63G/S107gA+ showed no significant effect (Figure 2b). These findings suggest that the double-nicking strategy is effective in eliminating HPV16-positive cells, though further optimization of the gRNA combination is needed.

### 2.3. Design of DN-gRNA Pairs Targeting the E6/E7 Region of HPV16

HPV16 E6 and E7 oncogenes are transcribed from the P97 promoter, and their inactivation is crucial for halting viral oncogene expression. To target these genes, we identified candidate gRNAs between nt. 7450 in the long control region (LCR) and nt. 1600 in the E1 ORF of the HPV16 genome, aiming to induce nicks with 5′-overhangs ranging from 26 to 134 nucleotides [7]. This search revealed 146 candidate gRNAs and 299 DN-gRNA pairs. Sequence alignment across ten sublineages (A1 to A5, B1 and B2, C and D1 and D2) [15,16] and the selection of 5′-overhangs between 30 and 60 nucleotides led to the identification of 57 gRNAs and 68 DN-gRNA pairs with minimal variation across sublineages (Appendix A). From these, we designed 24 gRNAs targeting 21 sites and 16 DN-gRNA pairs targeting the LCR/E6 boundary, the mid E6 region, the E6/E7 boundary, the mid E7 region, and the E1 region (Table 1). Lentiviral vectors expressing these DN-gRNAs were used to establish SiHa cell lines expressing one to four different sets of DN-gRNA pairs, followed by infection with Cas9n-expressing AdV, AxCB-Cas9n.

### 2.4. Growth Inhibition of SiHa Cells by DN-gRNA Pairs and Cas9n

The SiHa cell lines expressing DN-gRNAs (N1 to N6, B1 and B2, P1 to P3, H1 and H2) from the gRNA units introduced by lentivirus vectors (Figure 3a) were infected with an AdV expressing Cas9n. Among the DN-gRNAs, N3, B1, B2, P1, P2, P3, H1 and H2 significantly inhibited SiHa cell growth upon Cas9n infection, while N1, N2, N4, N5 and N6 had no significant effect (Figure 3b). Further testing of DN-gRNAs N1 to N7 in the HPV16-positive SCC90 oropharyngeal cancer cell line, which lacks the single-nucleotide variation (SNV) present in SiHa cells, revealed stronger growth inhibition with N3, N5, N6, and N7 than with N1, N2, and N4 (Figure 3c).

### 2.5. Growth Inhibition of SiHa Cells by Multiplex DN-gRNA Pair Combinations

Given the effectiveness of DN-gRNAs (N3, B1, B2, P1, P2, P3, H1, H2) in the SiHa cell lines, we tested combinations of up to four DN-gRNAs, including two additional DN-gRNAs (Z1 and Z2) targeting the E1 region (Figure 4a). More than 90% growth inhibition was observed in six of 22 combinations of two DN-gRNAs (Figure 4b, from B1Z2 to H2P1), 20 of 37 combinations of three DN-gRNAs (Figure 4c, from H1B1P1 to H1P1Z1), and 37 of 49 combinations of four DN-gRNAs (Figure 4d, up to N3P3B2Z1), but none of single sets of DN-gRNAs at an MOI of 250 (Figure 4a). These results indicate an additive or synergistic effect of multiplex DN-gRNAs. Considering that B1, B2, P1, P2, and H1 had no SNVs in any of the HPV16 sublineages, the combination of B1, P1, and H1 emerged as the most effective for targeting the E6/E7 region.

### 2.6. Growth Inhibition by Co-Infection of All-in-One AdVs Simultaneously Expressing Four DN-gRNA Pairs and AdV Expressing Cas9n

Using the “Tetraplex-guide Tandem” method to construct multiplex gRNA units [14], we developed AdVs expressing four DN-gRNA pairs targeting the HPV16 genome (N7, B1, P1, and H1, collectively termed Set2; vector designated Ax-Set2) or, as a control, DN-gRNAs specific for the hepatitis B virus (HBV) X gene (vector designated Axg8HBV-X) [17]. SiHa cells co-infected with Ax-Set2 and either AxCB-Cas9 or AxCB-Cas9n exhibited nearly complete cell death at an MOI of 1000 (Figure 5a,b). In contrast, cells co-infected with Axg8HBV-X showed only slight growth inhibition (Figure 5a,b, shown as Vec). These results highlight the specificity and efficacy of the HPV16-targeted Cas9n system.

To verify that the observed growth inhibition was due to the knockout of HPV16 E6 and E7, we established SiHa cells expressing HPV18 E6 and/or E7. While HPV18 E6 and E7 possess similar functions to HPV16 E6 and E7 in inactivating p53 and pRB, they are resistant to disruption by gRNAs targeting HPV16. Interestingly, the growth inhibition was partially rescued by the expression of HPV18 E6 alone, but not by HPV18 E7 alone, and was strongly rescued by the co-expression of HPV18 E6 and E7. These findings confirm that the growth inhibition induced by the HPV16-targeted Cas9n system is specifically dependent on the knockout of the E6/E7 genes.

### 2.7. Disruption of the HPV16 Genome by DN-gRNAs and Cas9n Expressed by All-in-One AdVs

We constructed an all-in-one AdV expressing four DN-gRNA pairs from Set2 targeting the HPV16 genome, along with Cas9n (Ax-Set2-PGKCas9n) (Figure 6a). SiHa cells, which contain a single copy of the HPV16 genome integrated into their genome, and SCC90 cells, which harbor more than 100 copies of the HPV16 genome integrated [18,19,20], as well as HCK1T cells, which are normal cervical keratinocytes immortalized by transduction of telomerase reverse transcriptase (TERT), were infected with Ax-Set2-PGK-Cas9n at MOIs ranging from 5 to 1500. In both SiHa and SCC90 cells, this AdV significantly inhibited cell growth at MOIs greater than 50, with complete inhibition at an MOI of 1500. In contrast, the growth of HCK1T cells was only minimally inhibited, even at an MOI of 1500 (Figure 6b).

### 2.8. Large Deletion of HPV16 Genome Induced by Four DN-gRNAs and Cas9n

To examine indels induced by multiplex gRNAs and Cas9 or Cas9n, we constructed AdV expressing four DN-gRNAs of Set2 (Ax-Set2). SiHa cells were infected with Ax-Set2 and either AxCB-Cas9 or AxCB-Cas9n at an MOI of 500. HPV16 DNA containing the target sites was amplified by PCR. The PCR products from SiHa cells infected with Ax-Set2 and AxCB-Cas9 formed smear bands around 0.7 kb, while those infected with Ax-Set2 and AxCB-Cas9n formed sharp bands around 0.7 kb. Mock-infected cells formed a sharp band around 1 kb, as expected (Figure 7a). The PCR products were subjected to long-read sequencing, and the sequences were aligned with the HPV16 genome (Figure 7b). More than 99% of the PCR products from Ax-Set2 and AxCB-Cas9- or AxCB-Cas9n-infected cells showed large deletions (Figure 7b, upper and middle), while more than 99% of those from mock-infected cells showed no deletions (Figure 7b, lower).

The majority of PCR products from Ax-Set2 and AxCB-Cas9 infected cells had precise large deletions between the break sites of S486G and S893G. However, due to an SNV in the AS443g+ target sequence of HPV16 DNA in SiHa cells of sublineage A5 (Appendix A), few products had deletions starting from the break site of AS443g+ (Figure 7b, upper). Most PCR products from Ax-Set2 and AxCB-Cas9n-infected cells displayed large deletions with heterogeneous ends, likely corresponding to target sites of B1 and H1. Few products had deletions starting from the putative N7 target sites (AS443g+/S486G) due to the SNV in the AS443g+ target sequence (Figure 7b, middle). Interestingly, most PCR products from Ax-Set2 and AxCB-Cas9-infected cells retained patchy segments (Figure 7b, upper, thin, light green lines) within the entire deletions, with their ends corresponding to the break sites of AS507G, S563g+, AS706g+, S753G, and AS859G. These complex remaining segments likely explain the smear bands around 0.7 kb. Similarly, the PCR products from Ax-Set2 and AxCB-Cas9n-infected cells showed large deletions between the DN break sites of B1, P1, and H1 (Figure 7b, middle). Some PCR products also contained patchy segments with heterogeneous ends (thin, pink lines). These results suggest that Cas9n can induce irreversible disruption of the E6/E7 genes by cooperatively introducing double-stranded DNA breaks at three DN-gRNA target sites (B1, P1, and H1), as efficiently as Cas9, which could induce double-stranded breaks at seven gRNA target sites.

### 2.9. Comparison of Off-Target Effects Induced by Cas9n and Wild-Type Cas9

A major obstacle to the clinical application of CRISPR/Cas9 gene editing technology is its off-target effects. The double-nicking technology has been shown to reduce these off-target effects [7]. To assess the extent of off-target effects induced by AdVs expressing Cas9 and multiplex gRNAs, and to evaluate the reduction in these effects by AdVs expressing Cas9n and multiplex gRNAs, we aimed to sequence putative off-target sites predicted by the CRISPRdirect [21]. We selected 10 genomic sites that matched at least 12 nucleotides at the 3′-end and had no more than 4 total mismatches (Appendix A).

HCK1T and SiHa cells were infected with Ax-Set2 and either AxCB-Cas9n or AxCB-Cas9, and genomic DNA was extracted two days post-infection. DNA segments containing the putative off-target sites were amplified by PCR, followed by ultra-deep sequencing of the PCR products using Illumina’s next-generation sequencer. The indel rate of the PCR products from mock-infected HCK1T Set2 cells was considered the background, representing errors due to sequencing or PCR, and ranged from 0.73% to 7.7% across the 10 putative off-target sites. Among these sites, the indel rate at eight sites ($1, $2, $4, $5, $6, $7, $8, and $10) in HCK1T and SiHa cells infected with Ax-Set2 and AxCB-Cas9n or AxCB-Cas9 was indistinguishable from the background indel rate observed in the mock-infected HCK1T cells (Appendix A). Interestingly, a marked increase in indels was observed at site $3 in SiHa cells infected with Ax-Set2 and AxCB-Cas9 at an MOI of 500. The background indel rate at site $3 was 0.85% (total reads = 97,600, indel reads = 825), while the indel rate in SiHa cells infected with Ax-Set2 and AxCB-Cas9 was 43.59% (total reads = 84,297, indel reads = 36,749). In contrast, the rate in SiHa cells with Ax-Set2 and AxCB-Cas9n was as low as 0.89% (total reads = 93,400, indel reads = 835) (Figure 7c; Appendix A). After subtracting the background indel rate, we estimated that the adjusted indel rate in the SiHa with Ax-Set2 and AxCB-Cas9n (0.049%) was approximately 860-fold lower than that in the cells with Ax-Set2 and AxCB-Cas9 (42.75%). Although the indel rate in HCK1T cells was lower than in SiHa cells, the rate at site $3 in HCK1T cells with Ax-Set2 and AxCB-Cas9 was 3.1% (total reads = 110,111, indel reads = 3467), and the rate in the cells with Ax-Set2 and AxCB-Cas9n was 0.9% (total reads = 102,102, indel reads = 914). Additionally, slight increases in the indel rate were detected at site $9 in SiHa and HCK1T cells with Ax-Set2 and AxCB-Cas9 (1.8% and 1.4%, respectively) compared to the background indel rate of 0.94%. However, the indel rate at $9 in SiHa and HCK1T cells with Ax-Set2 and AxCB-Cas9n was 1.1% and 1.2%, respectively. These data suggest that indels occurred much less frequently with AxCB-Cas9n compared to AxCB-Cas9 at the off-target sites.

### 2.10. Tumor Suppression In Vivo

We developed a new AdV expressing four DN-gRNAs (B1, P1, H1, and Z1, collectively referred to as SetA) and Cas9n driven by a truncated elongation factor 1αα (EFd) promoter (Ax-SetAEFdCas9n) (Figure 8a). The ability of this construct to inhibit SiHa cell growth was compared with Ax-Set2-PGKCas9n. Ax-SetAEFdCas9n was more than four times as effective as Ax-Set2PGKCas9n, as determined by the 50% inhibitory dose (Figure 8b). To evaluate the therapeutic potential, we tested SetA-EFdCas9n in xenograft models. Ax-SetA-EFdCas9n significantly inhibited the tumor growth of HPV16-positive SCC90 cells (Figure 8c). In a patient-derived xenograft model of HPV16-positive cervical cancer, three of six tumors were completely eradicated by Ax-SetA-EFdCas9n (Figure 8d).

## 3. Discussion

The progression of HPV-associated cancers and their precursor lesions is largely dependent on the expression of the viral oncogenes, E6 and E7. Given this, several strategies have been explored to suppress the expression of these viral oncogenes. Early studies demonstrated that the viral transcription factor, E2, could suppress E6/E7 transcription by binding to the viral promoter, thereby inducing apoptosis in HPV-infected cells [23].

Subsequently, RNA interference (RNAi) approaches, including small interfering RNA (siRNA) and short-hairpin RNA (shRNA), were developed to inhibit E6/E7 expression, showing promising in vitro results [24]. However, effective delivery systems for siRNA and shRNA were crucial for clinical application. Recently, adeno-associated virus (AAV) vectors expressing shRNAs targeting HPV16 E6/E7 have demonstrated tumor-suppressive effects in HPV16-positive cervical cancer cell lines [25]. Despite these advances, the reversible nature of RNA interference and the potential integration of AAV into the human genome present ongoing challenges.

With the advent of gene editing technologies, such as ZFN, TALEN, and CRISPR/Cas9, the potential for inducing irreversible disruption of the E6/E7 region of HPV genomes has emerged as a promising therapeutic avenue. Numerous studies have provided proof-of-concept that HPV genome editing could offer a potential cure for HPV-associated cancers [10,11,12,13,26,27,28,29,30,31,32,33,34,35,36,37,38,39,40]. Among these technologies, CRISPR/Cas9 has emerged as the most powerful and flexible gene-editing tool due to its relatively lower off-target effects compared to ZFN and TALEN [26,27]. For example, AAV vectors expressing Staphylococcus aureus Cas9 (saCas9) and two gRNAs targeting HPV18 E6 successfully inhibited HeLa cell xenograft growth [12]. Another approach is liposome delivery of CRISPR/Cas9 [41]. However, off-target activity remains a significant concern for the therapeutic application of CRISPR/Cas9 [42]. To address this, double-nicking (DN) technology was developed, which reduces off-target effects by 50- to 1500-fold in cell lines [7]. The main limitation of DN-technology is that it requires two gRNAs to induce a double-strand break, thereby increasing the complexity of the approach. However, the “Tetraplex-guide Tandem” method made it possible to construct cosmids and adenovirus vectors containing multiplex gRNA-expressing units [14]. In this study, using this approach, we developed an AdV expressing four DN-gRNA pairs (eight gRNAs) targeting the E6/E7 region of the HPV16 genome. Upon co-infection with an AdV expressing Cas9n, we observed significant cytocidal effects in HPV16-positive cancer cells. Specifically, the simultaneous expression of the four DN-gRNA pairs and Cas9n efficiently induced irreversible large deletions within the E6/E7 region of HPV16 DNA in SiHa cells, leading to extensive cell death. We also examined off-target mutations associated with the four DN-gRNA pairs using wild-type Cas9 and observed that one gRNA (S486G+) induced indels at a high frequency (43.59%) at a putative off-target site. However, when combined with Cas9n, the indel rate was dramatically reduced (0.89%, i.e., 1/110), approaching background error levels (0.85%).

To optimize the combination of multiplex gRNAs, we developed a method for constructing lentiviral vectors expressing DN-gRNA pairs with up to four different drug-selection markers. This enabled us to systematically evaluate the effects of various gRNA combinations on the growth of HPV16-positive SiHa cells. Our results showed that a single DN-gRNA pair moderately inhibited cell growth, while combinations of two or three pairs significantly enhanced growth suppression. This is particularly relevant given the diversity of HPV16 sublineages, which may harbor SNVs within the target sequences. Despite the presence of SNVs, our approach induced large deletions in the E6/E7 region, effectively killing SiHa cells. Analysis of 648 HPV16 strains from the NCBI database revealed that 89.8% and 96.0% of these strains can be cleaved at four sites by Set2 and SetA DN-gRNAs, respectively. These findings suggest that these gRNA sets may be effective across a broad population of HPV16-induced cancer patients (Appendix A).

Based on these findings, we attempted to construct an all-in-one AdV expressing four DN-gRNAs and Cas9n under a CB promoter. However, we encountered frequent loss of the units during propagation, likely due to high Cas9n expression levels. To resolve this, we constructed an AdV with a weaker PGK promoter, which allowed successful propagation without detectable unit deletion. The resultant Ax-Set2-PGKCas9n efficiently inhibited the growth of HPV16-positive cell lines, SiHa and SCC90, while sparing normal cervical epithelial cells (HCK1T). However, a higher multiplicity of infection (MOI) was required for optimal growth inhibition compared to a combination of Ax-Set2 and AdCB-Cas9n.

To improve efficacy, we designed a new all-in-one AdV expressing Cas9n under a truncated elongation factor 1α (EF1α) promoter, with gRNA pairs inserted into the E1 and E4 regions. The resulting Ax-SetA-EFdCas9n was easily propagated and exhibited more than fourfold greater inhibitory activity against SiHa cells in vitro compared to Ax-Set2-PGKCas9n.

All-in-one AdVs, which contain both multiplex gRNA-expression units and Cas9n expression units, are desirable because, in contrast to the co-expression strategy, the multiplex gRNA units and the Cas9n unit are always introduced simultaneously into a single cell. Very recently, Nakahara, T. et al. [9] reported that, although all-in-one AdVs containing four gRNA units can be constructed without unit deletion, constructing an all-in-one AdV containing eight gRNA units is difficult because the vector size exceeds the virus packaging limit. This problem was partly solved by further deletion of the L4/E3 region in the vector backbone. However, the efficiency of gene disruption was decreased, possibly due to the L4/E3 deletion. The all-in-one AdVs containing eight gRNA units used in this work overcame these two problems by further modification of the L4/E3 region, and the detailed structure will be reported elsewhere.

In a pre-clinical in vivo study, intratumoral injections of Ax-SetA-EFdCas9n into SCC90 xenografts and PDX-CC1 patient-derived xenografts significantly inhibited tumor growth, with some tumors completely regressing. However, challenges remain regarding the retention of viral fluid within tumors, particularly for larger or ulcerated tumors. Strategies to enhance the diffusion and retention of the viral fluid may improve therapeutic outcomes.

The efficiency of AdV-mediated gene editing primarily depends on the expression of the coxsackievirus and adenovirus receptor (CAR) on target cells, although other virus entry receptors have also been reported [43,44]. While CAR expression is elevated in many cervical cancers, approximately 20% of cases are CAR-negative [45]. In our study, SiHa cells exhibited high susceptibility to AdV-mediated gene editing despite being reported to lack detectable levels of CAR [46]. However, data from the Human Protein ATLAS (https://www.proteinatlas.org/, accessed on 24 December 2024) indicate that CAR is expressed in all eight cervical cancer cell lines tested, including SiHa. Given that CAR expression can decline in culture or in vivo, modifications to AdV—such as fiber switching to facilitate CAR-independent infectivity or using conditionally replicating AdV backbones—may be necessary to enhance gene transfer across a broader range of cancer cell types [47].

In this study, we established a novel adenoviral vector co-expressing Cas9n and DN-gRNAs against HPV16 E6/E7. Despite room for optimization of promoter usage and gRNA selection, the vector efficiently eliminated HPV16-positive cancer cells in vitro and in vivo while inducing minimal off-target mutations, a prerequisite for clinical translation. These findings provide strong proof of concept for a safe CRISPR-based therapy targeting HPV oncogenes. Because intratumoral delivery is required, this strategy is particularly suited for solitary HPV-positive oropharyngeal tumors, where repeated administration could achieve complete eradication. Since adenoviral E1A and E1B share functional similarities with HPV E7 and E6, respectively, in their ability to inactivate the tumor suppressors pRB and p53, it is feasible to design adenoviral vectors with enhanced replication specificity in HPV-positive cancer cells. Incorporating Cas9n and DN-gRNAs into such vectors may substantially enhance therapeutic efficacy.

## 4. Materials and Methods

### 4.1. Plasmids

The HPV18 E6E7, E7, and the splicing donor mutant E6SD segments were cloned into the destination vectors to generate pCLXSN-18E6E7, pCLXSN-18E7, pCLXSN-18E6SD, and pCMSCVbsd-18E6SD. Lentivirus vector expressing a red-shift mutant of click beetle luciferase (CBR2), CSII-CMV-CBR2-iresPuro, was constructed by recombining the CBR2 segment with CSII-CMV-RfA-ires-Puro by the LR reaction (Invitrogen, Waltham, MA, USA). CSII-CMV-RfA-ires-Puro (p6811) was constructed by combining an AgeI-HpaI segment of CSII-CMV-MCS kindly provided by Dr. Miyoshi (RIKEN BRC) with an AgeI-EcoRV segment of pDEST-PQCXIP. CBR2 segment was derived from ATG-2460 (Addgene plasmid # 108713; http://n2t.net/addgene:108713, accessed on 28 August 2025; RRID: Addgene_108713), which was a gift from Keith Wood [22].

Retrovirus vectors expressing a single gRNA and lentivirus vectors expressing DN-gRNAs were listed in Appendix A, respectively. The detailed construction of the vectors was described in Appendix A.

Briefly, self-inactivating retroviral vector plasmids with drug-resistant genes of either puro or bsd were used for expressing gRNAs targeting the HPV genome. gRNA expression cassette was originally derived from gRNA-GFP-T2, which was a gift from George Church (Addgene (Cambridge, MA, USA) plasmid # 41820; http://n2t.net/addgene:41820, accessed on 28 August 2025; RRID:Addgene_41820) [48]. The target sequence was mutagenized by PCR followed by an in-fusion reaction with the In-fusion HD kit (TAKARA Bio Inc., Kusatsu, Japan). Lentivirus vectors expressing DN-gRNAs were constructed by using the MultiSite Gateway Three-Fragment Vector Construction kit (Invitrogen). Entry clones, pENTR221-(L4-L1)-U6H1R-gRNA-GFP-T1 and pENTR221-(R4r-R3r)-U6H1R-gRNA-GFP-T2 were generated from gRNA_GFP-T2 (Addgene 41820). The target sequence was mutagenized by PCR followed by an in-fusion reaction with the In-fusion HD kit (TAKARA) (Appendix A). Successful mutagenesis was confirmed by Sanger sequencing. Five drug-resistant entry clones containing neo, puro, hyg, bsd, and zeo genes in the pENTR221(L3-L2) backbone were constructed. Three entry clones, one of the pENTR221(L1-L4)-gRNA plasmids, one of the pENTR221(R4r-R3r)-gRNA plasmids, and one of the pENTR221(L3-L2)-drug-resistant plasmids were recombined with a promoter-less lentivirus vector, CSII-RfA, by LR reaction according to the manufacturer’s instructions (ThermoFisher Scientific, Waltham, MA, USA) to generate lentivirus vectors expressing a pair of DN-gRNAs as well as a drug-resistant gene (Appendix A).

A lentivirus vector, ps4xlv-4shCas9-1723-2932-puro, expressing four shRNAs against Cas9 was developed using the Tetraplex-guide Tandem method [14]. The shRNA target sequences were 5′-gcgtggaagatcggttcaacg-3′, 5′-gagaatgaagcggatcgaaga-3′, 5′-gtccgatttccggaaggattt-3′, and 5′-ggggagatcgtgtgggataag-3′, all under the control of the U6 promoter. To construct the vector, a SwaI site was introduced to replace the EF1α promoter in pLVSIN-EF1α Pur (Takara Bio Inc.), and the fragment containing lentivirus vector sequences was inserted into a charomid vector [49] to create chLVpur-w. The four shRNA expression units were then tandemly connected using the Tetraplex-guide Tandem method [14] and inserted into the SwaI site, resulting in the production of chLVpur-4shCas9-1723-2932. Finally, a DNA fragment containing the lentivirus vector sequences with four shRNA expression units was excised from chLVpur-4shCas9-1723-2932 using NruI and self-ligated to produce ps4xlv-4shCas9-1723-2932-puro.

### 4.2. Retrovirus and Lentivirus Infection

Retroviruses and lentiviruses were prepared as described previously [50]. Briefly, the retroviral vector and packaging constructs, pCL-GagPol and pEF6/env (10A1), or the lentiviral vector and packaging constructs, pCAG-HIVgp and pCMV-VSV-G-RSV-Rev (gifts from Dr. Hiroyuki Miyoshi, RIKEN, BioResource Center (Tsukuba, Japan) were co-transfected into 293T cells using PEI (polyethylenimine)-max (Polysciences, Inc., Warrington, PA, USA) according to the manufacturer’s instructions, and the culture fluid was harvested 60 to 72 h post-transfection.

### 4.3. Cell Culture

HPV16-positive human cervical cancer cell line, SiHa (ATCC HTB-35), HEK293 (ATCC CRL-1573), and 293T cells were cultivated in high-glucose DMEM supplemented with 10% FBS. HPV16-positive human oropharyngeal cancer cell lines, UPCI:SCC90 (hereafter referred to as SCC90, ATCC CRL-3239), were cultivated in high glucose Y-27632, 500 nM A-83-01, 500 nM DMH-1. All the media were also supplemented with 100 µg/mL penicillin and 100 µg/mL streptomycin. All cell lines were maintained in an incubator with 5% CO_2_ at 37 °C. SiHa cells infected by lentiviruses or retroviruses with different drug-resistant genes, neo, puro, hyg, bsd, and zeo, were selected in the presence of 1 mg/mL G418, 1 µg/mL puromycin, 500 µg/mL hygromicin B, 50 µg/mL blasticidin S, and 1 mg/mL Zeocin, respectively. SCC90 cells infected by lentiviruses with puro and bsd genes were selected in the presence of 1 µg/mL puromycin and 20 µg/mL blasticidin S, respectively.

### 4.4. Colorimetric Cytotoxicity Assay

Cells were seeded on 96-well plates at a density of 1000 or 2000 cells/well. On the next day, every three wells were inoculated with AdVs at the indicated multiplicity of infection (MOI). When the cells in the mock-infected wells reached confluence at day 7 to 14, cells were fixed and stained with sulforhodamine B (SRB) (Sigma Aldrich, St. Louis, MO, USA), and the remaining cell numbers were estimated by the SRB assay [51].

### 4.5. Construction of Cosmids Containing AdV Genome Bearing Multiplex gRNA Units

Multiplex gRNA expression units targeting the HPV16 were constructed in accordance with the Tetraplex-guide Tandem method [14] using the oligonucleotides shown in Appendix A. The cosmids of psAxp4-HP6sg8DN2(4654-7388) and psAxp4-HP6sg8DNa(5474-88145)-oPGNC9, parent cosmids of AdVs Ax-Set2 and Ax-Set2-PGKCas9n, respectively, were constructed by inserting head and mid-block PCR fragments into the cassette cosmids of psAxp4-wpcp-g and psAxp4-pcp-oPGNC9 at the SwaI site and the PmeI site, respectively. The head-block reverse primer ampHeadTE-BxSs-R (5′-gcg AATATT CCACTCTTCTGG (BstXI) AGCT GTTGACGCCAGCAACTGTACA GGATCC-3′) and the mid-block forward primer ampMidTA-BxSs-F (5′-gcgAATATTG CCAGAAGAGTGG (BstXI) AGCT GTCAAC GGCGTCAGTTGCTG GCTAGC-3′) were used instead of ampl Head-A R and ampl Mid-A F. BstXI was used instead of AlwNI. For the construction of psAx4f-HP6sg8DNa(5474)-1(88145)oEFdNC9, parent cosmid of Ax-SetA-EFdCas9n, PCR products of head block and mid-block were digested with HincII, and were cloned into the PmeI and SwaI sites of psAx4f-pgpc-owEFdNC9, respectively.

### 4.6. Production of AdVs

To prevent the potential loss of gRNA units caused by Cas9 activity, HEK293 cells were first transduced by infection with a lentiviral vector, ps4xlv-4shCas9-1723-2932-puro, expressing four shRNAs against Cas9. The resulting cell line was subsequently infected with three lentiviral vectors: CSII-CMV-AcrIIA4-NLS-ires-Hyg, CSII-CMV-AcrIIA4-NLS-ires-Bsd, and CSII-CMV-AcrIIA4-NLS-ires-Zeo, which express the Cas9 inhibitor AcrIIA4 fused to the SV40 nuclear localization signal (NLS). The modified 293 cells were used for the production of Ax-Set2-PGKCas9n and Ax-SetA-EFdCas9n, which contain both multiplex gRNA expression units and Cas9 nickase expression units. The cells were transfected with the BstBI-linearized AdV genome in psAxp4-HP6sg8DNa(5474-88145)-oPGNC9 and psAx4f-HP6sg8DNa(5474)-1(88145)oEFdNC9 as previously described [9,14,17]. The modified HEK293 cells were also inoculated with the 1st to 5th generation of AdVs, and the cells were harvested at 3 days post-infection. For Ax-Set2 production, BstBI-linearized psAxp4-HP6sg8DN2(4654-7388) was transfected to HEK293 cells. The AxCB-Cas9, AxCB-Cas9n, the same AdVs as AxCBCas9 [14] and AxCBNC9 [14], respectively, and Axg8HBV-X [17] were provided by Juntendo University. The AxCBCas9 (RDB19290) and AxCBNC9 (RDB19291) are available from RIKEN Bioresource Center (RIKEN BRC: https://dna.brc.riken.jp/ja/gene_analysis/crispr_cas9#Cas9_AdV, accessed on 28 August 2025).

### 4.7. Purification of AdVs

AdVs were released from the cells by freezing and thawing 5 times, and the supernatants were frozen in liquid nitrogen and kept at −80 °C or directly purified by using an anion exchange membrane based on the protocol originally described for AAV purification [52]. Briefly, the supernatant was incubated with 12.5 U/mL of Benzonase (Merck, Darmstadt, Germany) at 37 °C for 30 min and filtered through 0.45 µm Millex-HV disk filters (Merck, Darmstadt, Germany). The crude lysate was mixed with 1/9th volume of binding buffer (500 mM HEPES pH 7.5, 2.1 M NaCl) and passed through two tandem Sartobind Q75 (Sartorius Japan K.K., Tokyo, Japan) disks, washed with washing buffer (50 mM HEPES pH 7.5, 350 mM NaCl), and eluted with 20 mL of 50 mM HEPES pH 7.5, 600 mM NaCl. The eluted fraction was concentrated by an Ultrafiltration filter (MW100,000) and equilibrated with 1× storage buffer (20 mM TrisHCl, pH 8, 25 mM NaCl, 2.5% glycerol).

### 4.8. Titration of AdVs

The AdV titer was quantified using quantitative real-time PCR as previously described [53], with some modifications. HeLa cells were utilized as recipient cells, and RNaseP primers (5′-AGGTTTGGACCTGCGAGCG-3′ and 5′-GAGCGGCTGTCTCCACAAGT-3′), along with an RNaseP probe (5′-HEX-TTCTGACCTGAAGGCTCTGCGCG-TAMRA-3′), were used as controls, replacing the previously employed β-actin primers and probe. The qPCR was performed using the ABI9500FAST system, following the manufacturer’s protocol: 50 °C for 2 min and 95 °C for 10 min, followed by 40 cycles of 95 °C for 15 s and 60 °C for 1 min (Applied BioSystems, Waltham, MA, USA).

### 4.9. In Vitro Cleavage of HPV16 DNA with Cas9, tracrRNA and Selected crRNAs

HPV16 DNA from CaSki cells was amplified by PCR using the primers 5′-GAATTCCActatttTGGAGGACTGGAATTTTGGTCTAC-3′ and 5′-AATCCCAtttctcTGGCCTTGTAATAAATAGCACATTC-3′. The resulting 3017 bp fragment was cloned into the EcoRV site of pBluescript II KS+, and the sequence was verified via Sanger sequencing. PCR products, amplified using the same primer set, were used as substrates. Cas9 protein was obtained from TAKARA, and the crRNA and tracrRNA were synthesized by Sigma. One to two pmoles of Cas9 protein (150–300 ng) were combined with 2 pmoles of tracrRNA and each crRNA, and incubated for 5 min at room temperature. The reaction mixture, containing 300 ng (0.15 pmol) of substrate DNA, 2 µL of 10× H buffer (TAKARA), and nuclease-free water to a total volume of 20 µL, was incubated at 37 °C for 60 min. The products were analyzed using 1% agarose gel electrophoresis and stained with ethidium bromide (EtBr).

### 4.10. Patient Specimens and PDX Establishment

Patient-derived xenograft (PDX) models were established from cervical cancer tumors obtained from a patient treated at the National Cancer Center Hospital, Japan. The samples were provided by the National Cancer Center J-PDX Library, Tokyo, Japan [54]. The protocol for using PDX-CC1 was approved by the institutional review board (NCCRI: 2015-123 approval date on 17 August 2015 and NCCRI: 2022-043 approval date on 29 July 2022), with written informed consent obtained from all patients. Tumor samples, ranging from 2 to 10 mm^3^, were immediately soaked in storage solution (Theliokeep, Bio Verde Inc., Kyoto, Japan) after collection and stored at 4 °C. The samples were transplanted subcutaneously into the flanks of 6-week-old female NOG mice (NOD.Cg-Prkdcscid Il2rgtm1Sug/ShiJic, In Vivo Science Inc., Kawasaki, Japan). Tumor growth and body weight were monitored weekly. All mice were euthanized by cervical dislocation under anesthesia, and tumors were excised. The PDX-CC1 PDX model was serially transplanted into immunodeficient mice, and the 5th generation tumors from Balb/c nude mice were used for subsequent experiments. Genomic DNA from the second generation of PDX-CC1 tumor cells was extracted and subjected to PCR with PGMY09/11 primers to amplify HPV L1 DNA, followed by reverse blot hybridization for HPV genotyping, as previously described [55]. Genomic DNA from organoids derived from the 3rd generation of PDX-CC1 tumor cells was also extracted and analyzed by PCR using various primer sets to detect HPV16 DNA. A 2 kb PCR product was successfully amplified using the primers 5′-CAGGAGCGACCCAGAAAGTTACC-3′ and 5′-ACCTCCATCATCTACCCTATC-3′, and subsequently confirmed by Sanger sequencing.

### 4.11. In Vivo Experiments

Four-week-old Balb/c nude mice were purchased from CLEA Japan, Inc. (Tokyo, Japan) for in vivo experiments. A total of 3 × 10^6^ CBR2 luciferase-expressing SCC90 cells (SCC90/CBR2) were injected subcutaneously under the dorsal skin of the mice. Tumor size was measured twice weekly using calipers, and tumor volume was calculated using the formula: length × (width^2^) × ½. Once tumors became palpable (mean tumor volume = 75 mm^3^) on day 21, purified Ax-SetA-EFdCas9n AdV or PBS was intratumorally injected on days 21, 22, and 23 of the first week, and on days 27, 28, 29, and 30 of the second week. In the third week, injections were performed daily from days 33 to 37. All mice were sacrificed by cervical dislocation when tumor volumes in the control group exceeded 1000 mm^3^ on day 41. Mice were administered D-luciferin (150 mg/kg, Promega, Madison, WI, USA) intraperitoneally at each interval, then anesthetized with 2% isoflurane, and imaged using the IVIS Lumina S5 system (open filter, binning factor = 2, field of view = 12.9 × 12.9 cm, f = 4, and 0.5 s exposure time). Tumors from the 4th generation of HPV16-positive PDX (PDX-CC1) models, approximately 10 mm^3^ in size, were subcutaneously transplanted under the dorsal skin of nude mice. Tumors became palpable at 12 out of 16 transplantation sites in four mice. A mouse, which developed two large tumors exceeding 1000 mm^3^ by day 72, was sacrificed and excluded from the analysis. When the mean tumor volume in the remaining three mice reached 250 mm^3^ on day 78, purified Ax-SetA-EFdCas9n AdV or PBS was injected intratumorally for three consecutive days per week until day 129. The remaining mice were sacrificed on day 134, as tumor volumes in the control group exceeded 1000 mm^3^. Body weight changes and injection site observations were recorded at the experimental endpoint. All mice were maintained under specific pathogen-free conditions, and the study protocol was approved by the National Cancer Center Ethics Committee in accordance with national and institutional guidelines for animal experimentation.

### 4.12. Selection of Putative Off-Target Sites

Potential off-target sites for 8 gRNAs were identified using CRISPRdirect (https://crispr.dbcls.jp/, accessed on 28 August 2025) [21]. Criteria for selection included a perfect match of 12 out of 19 target nucleotides at the 3′ end, the presence of an NGG PAM sequence, and no more than four mismatches in the remaining 7 nucleotides at the 5′ end. Ten putative off-target sites were amplified via PCR using primer sets (Appendix A). The PCR products (100 ng) were subjected to library preparation using the NEBNext Ultra II DNA Library Prep Kit (New England Biolabs, Ipswich, MA, USA). The libraries were pooled and sequenced with 150 bp paired-end reads on the MiSeq platform (Illumina, San Diego, CA, USA), according to the manufacturer’s instructions. The FASTQ files were generated using the Local Run Manager software (v4.1.0) (Illumina) and aligned to the GRCh38 reference to generate BAM files. After converting from BAM to SAM format using SAMtools (v1.13) [56], each pair of reads, R1 and R2, was connected by overlapping bases. The connected reads were re-aligned to the GRCh38 reference. Finally, reads with Cas9/Cas9n-mediated indels were extracted based on the Compact Ideosyncratic Gapped Alignment Report (CIGAR) strings from the BAM files.

### 4.13. Detection of Indels at On-Target Sites

SiHa cells were infected with Ax-Set2, along with either Ax-CB-Cas9n or Ax-CB-Cas9, at an MOI of 500. Genomic DNA was extracted two days post-infection. The genomic DNA was amplified by PCR using a set of primers, 5′-caggagcgacccagaaagttacc-3′ and 5′-tcactaagtggactacccaa-3′. The PCR products were subjected to long-read sequencing using the MinION platform (Oxford Nanopore Technologies, Oxford, UK). The sequencing library was prepared using a total of 1 μg of DNA and the Ligation Sequencing Kit (SQK-LSK110, Oxford Nanopore Technologies). The library was loaded onto a R9.4.1 flow cell (Oxford Nanopore Technologies) and sequenced for 2  h. The basecalling of the raw data was performed using Guppy (v5.0.11, Oxford Nanopore Technologies). Quality was assessed using MinIONQC (v1.4.1) [57]. Reads were aligned to the HPV16 genome (NCBI, KY549302) using Minimap2 (v2.17) [58]. Finally, the reads containing the primer sequences were extracted and visualized using SAMtools (v1.9) and the Integrative Genomics Viewer (IGV v2.17.4) [59], respectively.

## 5. Patents

PCT/JP2019/037255, priority to JP2019-121668, JP2018-17927 application filed by Institute of Microbial Chemistry, 24 September 2019, Novel virus vector and methods for producing and using same (Inventor: Tomoko Nakanishi, Izumu Saito); Worldwide application WO EP US JP.

PCT/JP2024/003376, priority to JP2023-014899 application filed by NCC, Juntendo University, Institute of Microbial Chemistry (joint application) 1 February 2024, Gene Therapy Pharmaceutical Composition for HPV-positive Cancer or HPV-positive Precancerous Lesions, and Recombinant Adenovirus Vector (Inventor: Tohru Kiyono, Tomomi Nakahara, Tomoko Nakanishi, Izumu Saito).

## Figures and Tables

**Figure 1 ijms-26-08685-f001:**
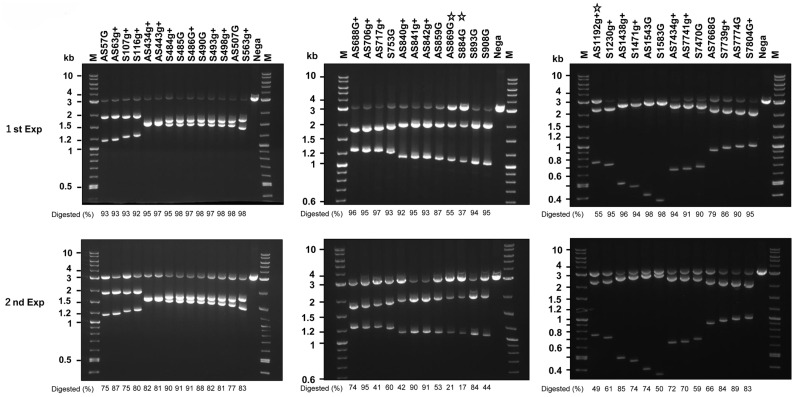
In vitro Cas9 activity with various crRNAs. The 3 kb HPV16 segments described in the Materials and Methods were digested with Cas9 mixed with tracrRNA and the indicated crRNA. Electropherograms of the digested products, along with the undigested substrates (Nega) and the molecular weight marker (M), are shown. Digestion efficiencies (%) are shown at the bottom of each lane. More than 50% of the substrate DNA that remained undigested by the crRNAs are marked with a star.

**Figure 2 ijms-26-08685-f002:**
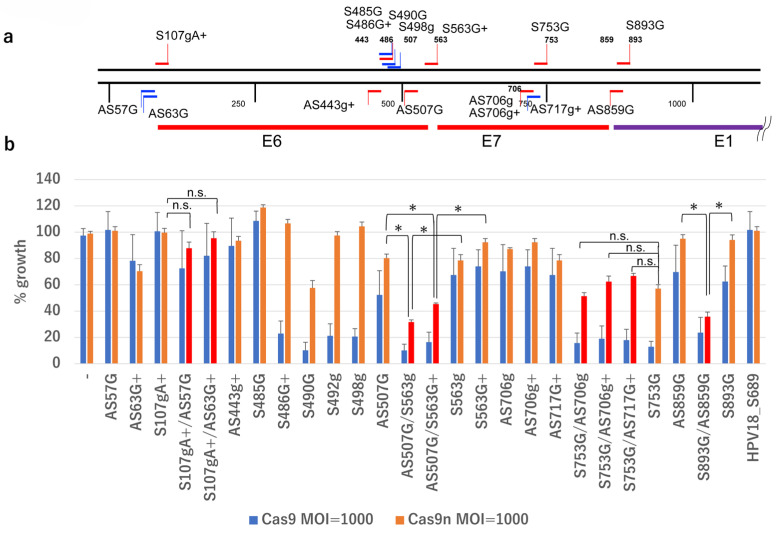
Growth inhibitory activity of representative gRNAs targeting HPV16 DNA. (**a**) Representative gRNAs on the map of the HPV16 genome, along with the coding sequences of E6, E7, and E1, are indicated. (**b**) Growth of SiHa cells expressing the indicated gRNA(s) and infected with AdV expressing Cas9 or Cas9n at an MOI of 1000 was measured. Statistical analysis was conducted using a *t*-test. Asterisks indicate a *p*-value < 0.05, and ‘n.s.’ indicates not significant.

**Figure 3 ijms-26-08685-f003:**
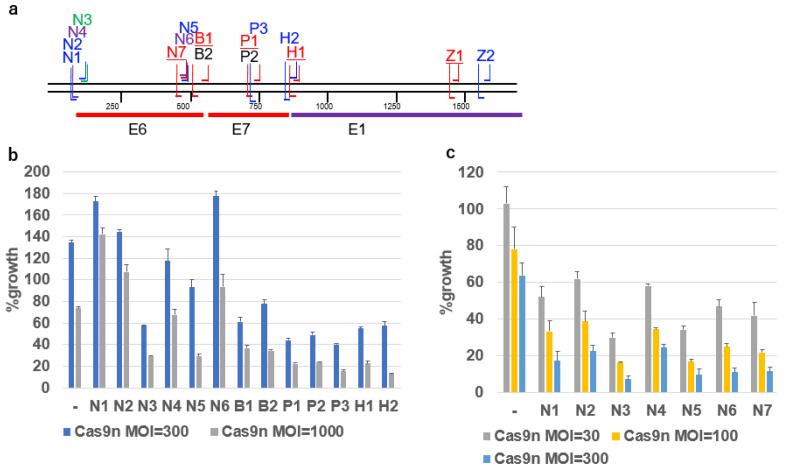
Growth inhibitory activity of representative DN-gRNAs targeting HPV16 DNA. (**a**) Schematic representation of representative DN-gRNAs on the map of the HPV16 genome, including the coding sequences of E6, E7, and E1. The gRNAs on the antisense strand for N5, N6, and N7 are common, and the gRNAs on the sense strand for P1, P2, and P3 are common. (**b**) Growth of SiHa cells expressing the indicated DN-gRNAs, infected with AdV expressing Cas9n at an MOI of 300 or 1000. The means of triplicate wells and error bars are shown. (**c**) Growth of SCC90 cells expressing the indicated DN-gRNAs, infected with AdV expressing Cas9n at an MOI of 30, 100, or 300. The means of triplicate wells and error bars are shown. In vitro Cas9 activity with various crRNAs.

**Figure 4 ijms-26-08685-f004:**
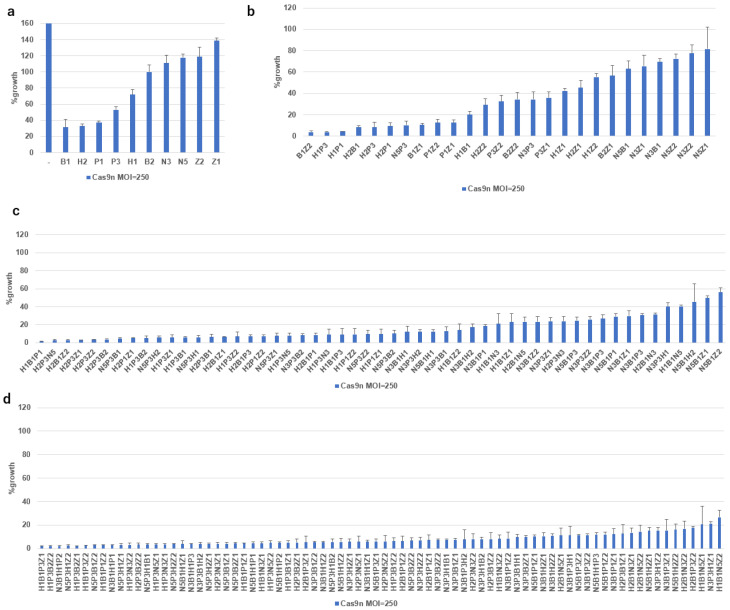
Growth inhibitory activity of representative DN-gRNAs targeting HPV16 DNA. Growth of SiHa cells expressing the indicated pair(s) of DN-gRNAs, infected with AdV expressing Cas9n at an MOI of 250. Growth of SiHa cells expressing a pair (**a**), two pairs (**b**), three pairs (**c**), or four pairs (**d**) of the indicated DN-gRNAs, infected with AdV expressing Cas9n at an MOI of 250, is presented in separate graphs, though all data were obtained in the same experiment. The means of triplicate wells and error bars are shown.

**Figure 5 ijms-26-08685-f005:**
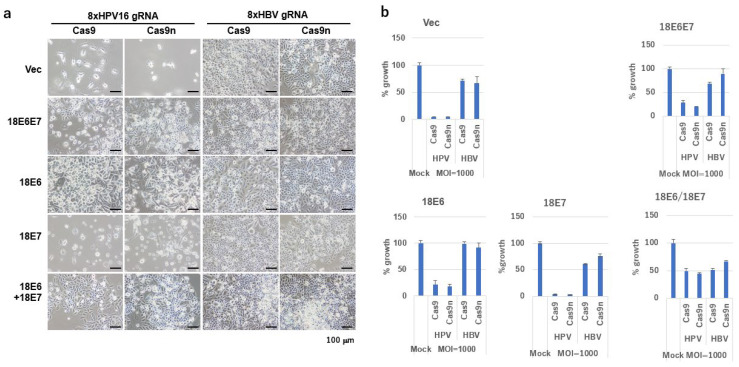
Growth inhibitory activity of Ax-Set2 and AxCB-Cas9 or AxCB-Cas9n on SiHa cells expressing HPV18 E6, E7, or E6E7. SiHa cells expressing HPV18 E6, E7, E6E7, 18E6 plus E7, or the LXSN vector were infected with Ax-Set2 (8xHPV16 gRNA), Axg8HBV-X (8xHBV gRNA)16 and AxCB-Cas9 or AxCB-Cas9n at an MOI. of 1000. (**a**) Representative images of SiHa cells 10 days post-infection are presented. Scale bars represent 100 μm. (**b**) Growth of SiHa cells 10 days post-infection was measured by the SRB assay and is shown in bar graphs. The means of triplicate wells and error bars are indicated.

**Figure 6 ijms-26-08685-f006:**
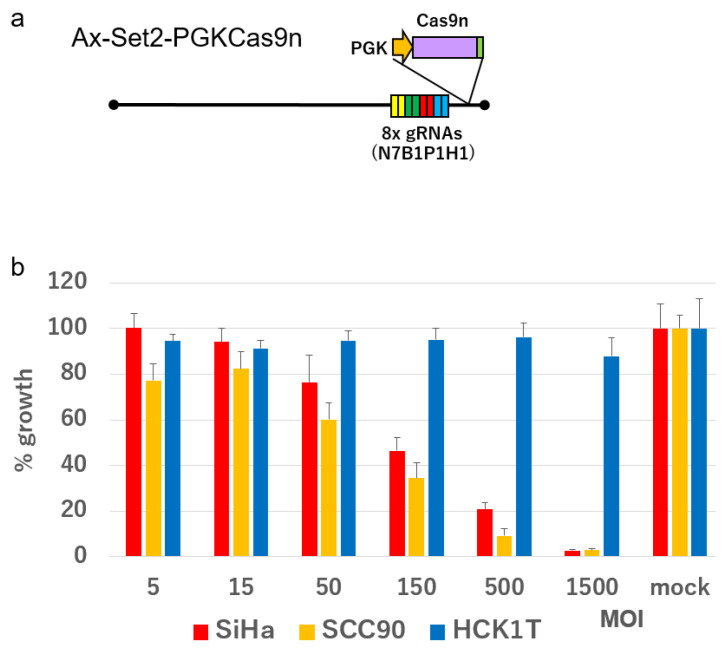
Growth of SiHa, SCC90, and HCK1T cells infected with Ax-Set2-PGKCas9n at the indicated MOI. (**a**) Schematic structure of Ax-Set2-PGKCas9n. (**b**) SiHa, SCC90, and HCK1T cells were infected with Ax-Set2-PGKCas9n at the indicated MOI. Growth of the cells was measured by the SRB assay and is shown in bar graphs. The means of triplicate wells and error bars are indicated. The means of mock-infected cells were set at 100%.

**Figure 7 ijms-26-08685-f007:**
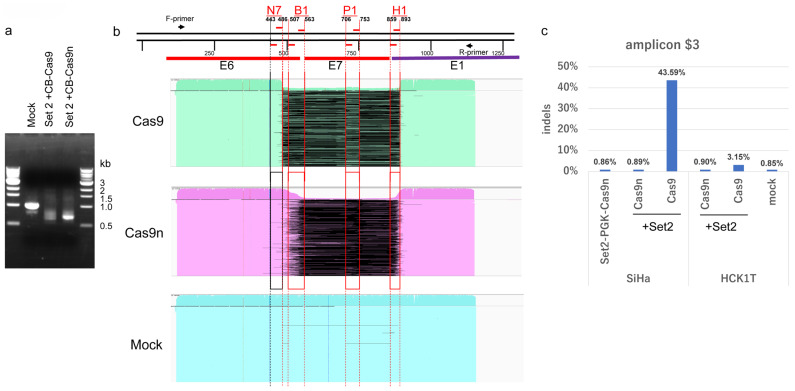
Indels in HPV16 DNA in SiHa cells at on-target sites and an off-target site induced by infection of Ax-Set2 and Ax-CB-Cas9 or Ax-CB-Cas9n. (**a**) Electropherogram of the PCR products of HPV16 DNA from SiHa cells at two days post-infection with the indicated adenoviruses. (**b**) Alignment of long-read sequences of the PCR products with the HPV16 genome. Sequences from Cas9-infected cells (light green), Cas9n-infected cells (pink), and mock-infected cells (light blue) are illustrated, with deleted sequences shown in black. DN-gRNAs expressed by Set 2 AdV and the positions of putative double-stranded break sites by Cas9 or putative nicking sites by Cas9n are indicated by red lines. As the HPV16 genome in SiHa cells has an SNV at nt. 442, the vertical line at nt. 443 is indicated by a black line. Depth of coverage is shown at the top of each panel. (**c**) Frequencies of indels induced in a putative off-target site. From left to right: SiHa cells infected with Ax-Set2-PGKCas9n; SiHa cells infected with Ax-Set2 and Ax-CB-Cas9n or Ax-CB-Cas9; HCK1T cells infected with Ax-Set2 and Ax-CB-Cas9n or Ax-CB-Cas9; mock-infected HCK1T cells. See Materials and Methods for details.

**Figure 8 ijms-26-08685-f008:**
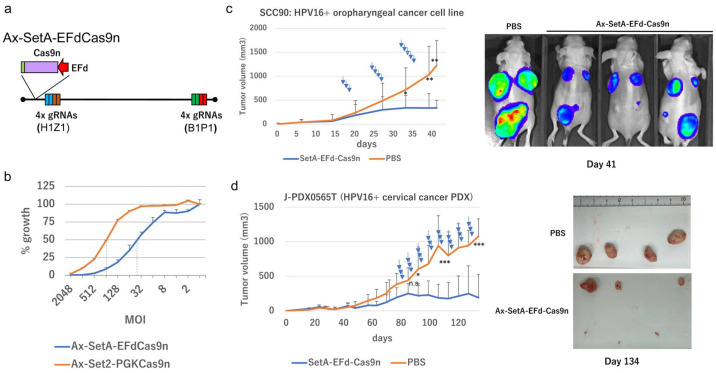
Growth inhibition of HPV16-positive cells and tumors by “All-in-One” AdV targeting HPV16. (**a**) Schematic structure of Ax-SetA-EFdCas9n. (**b**) Growth of SiHa cells infected with Ax-Set2-PGKCas9n or Ax-SetA-EFdCas9n. Growth of SiHa cells infected with Ax-Set2-PGKCas9n (red line) or Ax-SetA-EFdCas9n (blue line) at the indicated MOI. The means of triplicated wells and error bars are plotted. (**c**) Tumor growth of SCC90 cells expressing Click beetle luciferase mutant (CBR2). A total of 3 × 10^6^ SCC90 cells expressing CBR2 luciferase [22] were injected under the dorsal skin of nude mice on day 0. Ax-SetA-EFdCas9n AdV or PBS was intratumorally injected 12 times, as indicated by the blue arrows. Depending on the batch of purified AdVs, different amounts of AdV (0.4 to 4 × 10^10^ pfu/site) were injected on the following days: day 21, 22, and 23 (2.1 × 10^10^ pfu/site); day 27, 28, and 29 (2.4 × 10^10^ pfu/site); day 30 (1.2 × 10^10^ pfu/site); day 33 (2.3 × 10^10^ pfu/site); day 34 (4.0 × 10^10^ pfu/site); day 35 (4.0 × 10^9^ pfu/site); and day 36 and 37 (2.5 × 10^10^ pfu/site). IVIS images were taken on day 41, as described in the Materials and Methods. (**d**) Tumor growth of PDX-CC1 tumors. Pieces of the 4th generation of an HPV16-positive PDX-CC1 tumor (approximately 10 mm^3^) were subcutaneously transplanted under the dorsal skin of nude mice on day 0. Ax-SetA-EFdCas9n AdV (2 × 10^10^ pfu/site) or PBS was intratumorally injected once a day for three consecutive days per week from day 78 to day 129, as indicated by the blue arrows. Photographs of excised tumors on day 134 are presented. Note that the largest tumor in the experimental group had a couple of ulcers, and the majority of the injected AdV fluid leaked through the ulcers. The statistical analysis was conducted using a *t*-test. Asterisks: *** indicates *p*-value < 0.01, ** indicates *p*-value < 0.02, and * indicates *p*-value < 0.05. “n.s.” indicates not significant (*p*-value > 0.05).

**Table 1 ijms-26-08685-t001:** Target sequences of DN-gRNAs and sublineages with single-nucleotide variations (SNV).

Name	Sense	Target Sequence (5′ to 3′)	Anti-Sense	Target Sequence (5′ to 3′)	SNV
N1	16_S107gA	gAAAAGAGAACTGCAATGTTTC	16_AS57G	GCTTTTATACTAACCGGTTT	C
N2	16_S107g+	gAAAGAGAACTGCAATGTTTC	16_AS57G	GCTTTTATACTAACCGGTTT	C
N3	16_S116g+	gTGCAATGTTTCAGGACCCAC	16_AS57G	GCTTTTATACTAACCGGTTT	C
N4	16_S107g+	gAAAGAGAACTGCAATGTTTC	16_AS63g+	gATGTCTGCTTTTATACTAAC	C
N5	16_S490G	GATTCCATAATATAAGGGGT	16_AS443g+	gAGATGTCTTTGCTTTTCTTC	A5
N6	16_S485G	GCAAAGATTCCATAATATAA	16_AS443g+	gAGATGTCTTTGCTTTTCTTC	A5
N7	16_S486G	GCAAAGATTCCATAATATAAG	16_AS443g+	gAGATGTCTTTGCTTTTCTTC	A5
B1	16_S563g+	gAACCCAGCTGTAATCATGCA	16_AS507G	GCAACAAGACATACATCGAC	-
B2	16_S563g	gACCCAGCTGTAATCATGCA	16_AS507G	GCAACAAGACATACATCGAC	-
P1	16_S753G	GCAAGTGTGACTCTACGCTT	16_AS706g+	gATATTGTAATGGGCTCTGTC	-
P2	16_S753G	GCAAGTGTGACTCTACGCTT	16_AS706g	gTATTGTAATGGGCTCTGTC	-
P3	16_S753G	GCAAGTGTGACTCTACGCTT	16_AS717g+	gAAAAGGTTACAATATTGTAA	D2, D3
H1	16_S893G	GCAGGTACCAATGGGGAAGA	16_AS859G	GGATCAGCCATGGTAGATTA	-
H2	16_S884G	GCTGATCCTGCAGGTACCAA	16_AS840g+	gATGGTTTCTGAGAACAGATG	A4
Z1	16_S1471g+	gACTAAAAACTAGTAATGCAA	16_AS1438g+	gACATTTAAAATATTTGTAAG	C
Z2	16_S1583G	GATTGGTGTATTGCTGCATT	16_AS1543G	GTTGATTTATTACTTTTAAA	-

## Data Availability

The original contributions presented in this study are included in the article/Appendix A. The raw data for the figures can be found in the Appendix A. Further inquiries can be directed to the corresponding authors.

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
