# Peer review of "Disruption of Human Papillomavirus 16 E6/E7 Genes Using All-in-One Adenovirus Vectors Expressing Eight Double-Nicking Guide RNAs"

_ijms, 2025, doi:10.3390/ijms26178685_

Round 1
Reviewer 1 Report
Comments and Suggestions for Authors
In this manuscript, authors selected optimal combinations of gRNAs targeting the HPV16 genome to effectively kill HPV16-positive cancer cells in conjunction with Cas9n by using CRISPR/Cas9 technology, they utilized a double-nicking technique to introduce DNA breaks in the E6 and E7 regions of HPV16 for overcoming the challenge in CRISPR/Cas9 therapy off-target effects. An "all-in-one" adenovirus (AdV) system expressing four gRNA pairs and Cas9n showed promise in inhibiting tumor growth in HPV16-positive cancer models, demonstrating its potential as a safe and effective treatment for HPV-induced tumors. The double-nicking strategy reduced off-target insert-deletion formation to 1/860 compared to wild-type Cas9. This study provides proof of concept that the combination of Cas9n and multiplex DN-gRNAs may offer a safe and effective therapeutic strategy for HPV-positive cancers. This is an overall interesting and technically well performed study. However, there are several pitfalls that the authors should be addressed.
- This study introduced DNA breaks in the E6 and E7 regions of HPV16 using the double-nicking technique, which only covered the replication of one high-risk HPV type. Can more gRNAs be designed to cover multiple serotypes of high-risk HPV? This would be more meaningful.
- The authors used "all-in-one" adenovirus to express gRNA pairs and Cas9n, and also delivered various combinations of double nicking (DN)-gRNA pairs to HPV16-positive cells via lentivirus, followed by Cas9 nickase (Cas9n) expression. Why were two different viral vectors chosen? Does lentivirus have the same risk of insertional mutagenesis as HPV?
- The control gRNA sequence for HPV18 is missing in Table 1.
- In Figures 3 and 4, the infection intensities of AdV varied greatly. How were these MOI values determined? Would maintaining consistency in MOI be more conducive to conclusion judgment?
- Lines 203-210, regarding the experiments on HPV18 E6/E7, no reference to the data is provided in this paragraph. Is it in Figure 5a-b? Should an additional 5c data be added?
- Some figures represent very simple data, such as Figures 6, 7, and 8. Many figures can be classified and combined according to scientific findings. I suggest the number of in-text figures should be controlled within six.
Check grammar and spelling mistakes thoroughly.
Author Response
Thank you very much for taking the time to review this manuscript. Please find the detailed responses below and the corresponding revisions/corrections highlighted/in track changes in the re-submitted files.

Reviewer 2 Report
Comments and Suggestions for Authors
In the manuscript, the authors attempted to develop an approach to gene editing using Cas9n that would be more specific than the approach using traditional Cas9. The manuscript is difficult to read because each section does not contain clear conclusions or leaders that should be used in the next section. For example, different sets of gRNAs pairs are used in different paragraphs. For some, specificity is assessed, while others are used in the xenograft model. The author needs to modify the manuscript so that it is possible to evaluate the properties of the selected sets of gRNA pairs from in vitro (activity, specificity) to in vivo (tumor volume). After taking into account the reviewer's suggestions, the manuscript can be published in IJMS MDPI.
Main
- The goal was to develop an approach based on Cas9n with fewer off-target effects then Cas9. According to Figure 5, Cas9n provides little or no increase in specificity compared to Cas9. Please, explain
- Please, include in the introduction a brief description of immunotherapeutic and other approaches using, for example, intrabodies and small molecule drugs [https://doi.org/10.1186/s13027-025-00641-7]. Also discuss an alternative CRISPR/Cas9-based approach with E6/E7 deletion in cervical cancer cells [https://doi.org/10.1002/jmv.28144] and compare the pros and cons (this and yours).
- Please explain why you did not use N7 (Table 1, Figure 3a) in SiHa cell lines, but only in SCC90 oropharyngeal cancer cell line (paragraph 2.4). In addition, the authors used N7 in “Tetraplex-guide Tandem” (paragraph 2.6), but it is not previously studied either separately (Figure 3) or in combination (Figure 4). Why was the set2 (N7, B1, P1 and H1) chosen in the paragraph 2.6 if such a combination was not studied in the paragraph 2.5? Then, in paragraph 2.10 DN-gRNAs (B1, P1, H1 and Z1, SetA) was used. Why is the specificity not checked for SetA in the paragraph 2.9 (Cas9 vs. Cas9n)?
- Lines 158-161: «Further testing of DN-gRNAs N5, N6, and N7 in the HPV16-positive SCC90 oropharyngeal cancer cell line, which lacks the single nucleotide variation (SNV) present in SiHa cells, revealed moderate growth inhibition with N3, N5, N6 and N7 (Figure 3c).»
In fact, at Cas9n MOI 300, N3, N5, and N6 revealed higher inhibition of SCC90 growth then N3, B1, B2, P1, P2, P3, H1 and H2 in SiHa cells (compare 3b and 3C). So, is it moderate growth inhibition?
- Lines 385-386: It is strange to read that: «The main limitation of DN-technology is that it requires two gRNAs to induce a double-strand break, thereby increasing the complexity of the approach.» and in the next paragraph that «tetraplex-guide Tandem” method to construct cosmids containing multiplex gRNA-expressing units (EIGHT gRNAs, more complex system). Please, modify the text.
- Add a paragraph to the discussion section that summarizes the results obtained and the range of their possible applications.
Typos:
- lines 59-61: «Although the neoplastic progression of HPV-associated cancers and their precursor lesions is known to depend on the expression of the viral oncogenes, E6 and E7 [2, 3], no targeted therapies for HPV have been developed»
There are 2 questions:
- a) Although the neoplastic progression of HPV-associated cancers and their precursor lesions ARE known to depend?
- b) Since neoplastic progression is the process by which normal tissue/benign tumor turn into malignant tumor, is it correct to write «the neoplastic progression of HPV-associated cancers»? Cancer is already a malignant tumor.
- lines 95-96: «This section may be divided by subheadings. It should provide a concise and precise 95 description of the experimental results, their interpretation, as well as the experimental 96 conclusions that can be drawn». Please delete
- Line 108: The 3 (according to Mat and Met) kb HPV16 segments
- Line 142: please decipher the abbreviation LCR
- Line 158: Further testing of DN-gRNAs N1-N7 …..?
- Lines 182-84: «Considering that B1, B2, P1, P2 and H1 had no SNVs 182 in any of the HPV16 sublineages, the combination of B1, P1, and H1, which, emerged as the most effective for targeting the E6/E7 region.» Please, rephrase grammatically correctly
- Lines 195-198: «Using the “Tetraplex-guide Tandem” method to construct multiplex gRNA units[13], we developed AdVs expressing four DN-gRNA pairs (N7, B1, P1 and H1, collectively termed Set2) targeting the HPV16 genome (Ax-Set2) or hepatitis Ba virus (HBV) X gene (Axg8HBV-X)[16] as a control.»
Please clarify if N7, B1, P1 and H1 actully have targets in hepatitis B virus (HBV) X gene? if not then rephrase
- Lines 198-200: «SiHa cells co-infected with Ax-Set2 and either AxCB-Cas9 or AxCB-Cas9n exhibited strong growth inhibition at an MOI of 300 and nearly complete cell death at an MOI of 1,000(Figure 5a,b).» The reviewer did not find MOI 300 in Figure 5a,b.
- Lines 288-290: We selected 10 genomic sites that matched at least 12 nucleotides at the 3'-end and had fewer than 3 total mismatches (Supplementary Table S2). In Table S2 «less than 4 mismatches in the rest of 7 nucleotides in the 5’-region»
- Line 472: which was a gift from
- HEK293T cells (line 507) or HEK293 (lines 511-512) were used?
Author Response
Thank you very much for taking the time to review this manuscript. We are encouraged by your positive comments. Please find the detailed responses below and the corresponding revisions/corrections highlighted/in track changes in the re-submitted files.

Round 2
Reviewer 1 Report
Comments and Suggestions for Authors
The manuscript has been thoroughly revised and I have no further comments.
Reviewer 2 Report
Comments and Suggestions for Authors
The authors took into account all the recommendations of the reviewer